# Temporal Adaptive Convolutional Intervention Network for Counterfactual Estimation: A Domain Generalization Perspective

## Abstract

Accurate estimation of time-varying treatment effects is crucial for optimizing interventions in personalized medicine. However, observational data often contains complex confounding bias and temporal complexities, making counterfactual estimation challenging. We propose Temporal Adaptive Convolutional Intervention Network (TACIN), a novel model that introduces an Intervention-aware Functional Convolution kernel to emphasize the role of treatments and capture complex temporal treatment interactions. TACIN addresses confounding bias from a domain generalization perspective, approximating the unknown target domain using adversarial examples and incorporating Sharpness-Aware Minimization to derive a generalization bound. This approach is more suitable for longitudinal settings compared to existing methods inspired by domain adaptation techniques due to inherent differences between static and longitudinal contexts. Experiments on simulated datasets demonstrate TACIN's superior performance compared to state-of-the-art models for counterfactual estimation over time. The code for reproducing the experimental results is available in an anonymous repository at `https://anonymous.4open.science/r/TACIN-2D20`.

## 1 Introduction

Precise estimation of time-varying treatment effects is crucial in domains like personalized medicine, where optimizing individual interventions relies on accurate estimating counterfactual outcomes (Huang & Ning, 2012). Randomized Controlled Trials, despite being the gold standard for causal inference (Hariton & Locascio, 2018), are limited by high costs and ethical concerns. Consequently, research has shifted focus to methods tackling confounding bias and temporal complexities in observational data for accurate counterfactual estimation.

Time-varying confounders often introduce complex confounding bias in observational data, leading to inaccurate estimates. Recent studies (Bica et al., 2020b; Melnychuk et al., 2022; Wang et al., 2024) have addressed this issue by learning representations to break the association between historical information and treatment assignment, drawing inspiration from domain adaptation techniques used in static causal inference settings (Shalit et al., 2017; Hassanpour & Greiner, 2019; Johansson et al., 2022). However, the suitability of domain adaptation in longitudinal settings remains questionable due to inherent differences between the two contexts.

Due to the unobservability of counterfactuals, we aim to train a model on the factual (source) domain that generalizes well to the counterfactual (target) domain. In static settings (Figure 1(a)), the target domain during testing is consistent with training and can be sampled, aligning with domain adaptation (Ganin et al., 2016). However, in longitudinal settings, the historical information of length $t$ generated under the training intervention policy (Figure 1(b)) differs from the $\tau$-step intervened historical information under a different testing policy (Figure 1(c)). The unknown true data generation process renders the target distribution unknown during testing, potentially explaining the ineffectiveness of current balancing strategies in longitudinal settings (Huang et al., 2024).

Another challenge in longitudinal settings arises from temporal complexities, particularly those caused by interactions between treatments over time. For instance, Roemhild et al. (2022) emphasize the existence of complex temporal interactions between antibiotics, which is crucial for

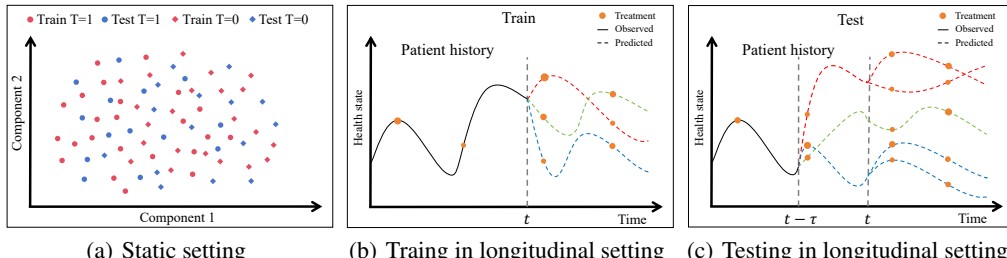

Figure 1: In static settings, the allocation of treatments does not affect the generation of covariates. As illustrated in Figure 1(a), confounding bias leads to distributional differences among intervention groups, while the distribution within each treatment group remains consistent across training and testing sets. However, in longitudinal settings, the data distribution is determined by a sequence of treatments. When the intervention policy during the testing phase from time $t - \tau$ to $t$ (Figure 1(c)) differs from that of the training phase (Figure 1(b)), it is infeasible to sample from the true distribution, as the data generation mechanism is unknown.

optimizing antibiotic use and minimizing resistance. To address this, Wang et al. (2024) propose a dual-module architecture called ACTIN, which highlights these interactions by processing historical and current treatment information at the same scale. However, similar to previous works (Bica et al., 2020b; Melnychuk et al., 2022), ACTIN treats treatments as general inputs, overlooking the unique role that treatments play. This oversight may hinder the model's ability to capture complex temporal treatment interactions. In static settings, methods such as VCNet (Nie et al., 2020) have demonstrated that emphasizing the role of treatments is a key point in model design, but in longitudinal settings, this remains an open question.

To address these challenges, we propose a novel model called Temporal Adaptive Convolutional Intervention Network (TACIN) for counterfactual estimation over time. In order to emphasize the role that treatments play, we draw inspiration from VCNet, which allows the weights of the prediction head to be functions of the treatment. TACIN introduces an Intervention-aware [1] Functional Convolution (IFC) kernel, ensuring that the convolutional kernel at each time step depends on the related treatment. Specifically, we employ Radial Basis Functions (RBFs) as basis functions to fit nonlinear functions of treatments, enhancing the model's ability to handle complex temporal treatment interactions. Furthermore, since the target domain is unobservable in longitudinal settings, we revisit confounding bias from the perspective of domain generalization. TACIN approximates the unknown target domain by generating adversarial examples using the Fast Gradient Sign Method (FGSM) (Goodfellow et al., 2015). By incorporating Sharpness-Aware Minimization (SAM) (Foret et al., 2021), a technique for enhancing model generalization performance, we derive a generalization bound for the model under certain conditions. The objective function proposed based on this theoretical analysis can more effectively mitigate time-varying confounding bias. A comprehensive review of related work, which motivates and contextualizes our contributions, is provided in Appendix A, which motivates and contextualizes our contributions. Extensive experiments on simulated datasets validate the effectiveness of TACIN, demonstrating superior performance over state-of-the-art models for counterfactual estimation over time.

## 2 PROBLEM FORMULATION

Consider an i.i.d. observational dataset $\mathcal{D}$ containing detailed information for $N$ patients, denoted as $\mathcal{D} = \left\{ \mathbf{x}_t^{(i)}, \mathbf{a}_t^{(i)}, \mathbf{y}_t^{(i)} \,_{t=1}^{T^{(i)}} \cup \mathbf{v}^{(i)} \right\}_{i=1}^{N}$. For each patient $i$, we observe time-varying covariates $\mathbf{X}_t^{(i)} \in \mathcal{X}$, treatments $\mathbf{A}_t^{(i)} \in \mathcal{A}$, and outcomes $\mathbf{Y}_t^{(i)} \in \mathcal{Y}$ at discrete time steps $T^{(i)}$, along with static covariates $\mathbf{V}^{(i)} \in \mathcal{V}$ such as gender and age. For notational simplicity, we omit the patient-specific superscript $(i)$ when the context is clear.

---

[1] In causal inference, "intervention" is a broad term. We primarily use "treatment" for medical interventions, although the terms are used interchangeably herein.

Building upon the potential outcome framework (Rubin, 1978) and its extension to time-varying treatments (Robins & Hernan, 2008), we aim to estimate time-varying counterfactual outcomes (Bica et al., 2020b; Lim et al., 2018; Li et al., 2021). Let $\bar{\mathbf{H}}_t = (\bar{\mathbf{X}}_t, \bar{\mathbf{A}}_{t-1}, \bar{\mathbf{Y}}_t, \mathbf{V})$ denote the patient's historical information, where $\bar{\mathbf{X}}_t = (\mathbf{X}_1, \cdots, \mathbf{X}_t)$, $\bar{\mathbf{Y}}_t = (\mathbf{Y}_1, \cdots, \mathbf{Y}_t)$, and $\bar{\mathbf{A}}_{t-1} = (\mathbf{A}_1, \cdots, \mathbf{A}_{t-1})$. Our objective is to estimate the potential outcome $\mathbf{Y}_{t+\tau}[\bar{\mathbf{a}}_{t:t+\tau-1}]$ following a treatment sequence $\bar{\mathbf{a}}_{t:t+\tau-1} = (\mathbf{a}_t, \cdots, \mathbf{a}_{t+\tau-1})$, conditioned on the historical information $\bar{\mathbf{H}}_t$:

$$\mathbb{E}[\mathbf{Y}_{t+\tau}[\bar{\mathbf{a}}_{t:t+\tau-1}]|\bar{\mathbf{H}}_t]. \tag{1}$$

To identify treatment effects from observational data, we rely on the assumptions of consistency, sequential ignorability, and sequential overlap, as established in prior works (Lim et al., 2018; Bica et al., 2020b; Melnychuk et al., 2022; Wang et al., 2024) and detailed in Appendix B.

## 3 METHOD

### 3.1 INTERVENTION-AWARE FUNCTIONAL CONVOLUTION

To enhance the model's ability to emphasize the role of treatments and capture potentially complex temporal interactions, we propose an Intervention-aware Functional Convolution (IFC) kernel. Explicitly encoding treatment information within the functional convolution operation enables IFC to better understand treatment impact on patient outcomes over time. This intervention-aware approach facilitates the learning of intricate relationships between treatments and covariates, ultimately improving the model's capacity for predicting counterfactual outcomes accurately.

In our proposed IFC, we employ a one-dimensional dilated convolution kernel that differs from traditional approaches (Oord et al., 2016) by incorporating the corresponding treatment as a function at each time step. Specifically, the output at time step $t$ can be computed as:

$$F(t) = \sum_{i=0}^{k-1} \mathbf{W}_i \phi(\mathbf{a}_t) \mathbf{z}_{t-d \cdot i} + \mathbf{b}. \tag{2}$$

In the equation, $F(t) \in \mathbb{R}^{d_{\text{out}}}$ denotes the output at time step $t$, where $d_{\text{out}}$ is the output dimension. The convolution kernel size is denoted by $k$, while $d$ denotes the dilation factor. $\mathbf{W}_i \in \mathbb{R}^{d_{\text{out}} \times d_{\text{in}}}$ is the $i$-th convolution weight matrix, where $d_{\text{in}}$ is the input dimension. The intervention function $\phi(\mathbf{a}_t)$ takes the treatment $\mathbf{a}_t$ at time step $t$ as input. $\mathbf{z}_{t-i} \in \mathbb{R}^{d_{\text{in}}}$ is the input at time step $t - i$, and $\mathbf{b} \in \mathbb{R}^{d_{\text{out}}}$ denotes the bias.

Radial Basis Functions (RBFs) are widely used as basis functions for approximating nonlinear functions and capturing complex interactions between variables. In this study, we explore two types of RBFs, Gaussian and Multiquadric, as the basis functions in the IFC kernel to capture the temporal treatment interactions.

Let $\mathbf{c} \in \mathbb{R}^{d_c}$ denote the centers of the RBFs. The Gaussian RBF for the $i$-th dimension of the treatment $\mathbf{a}$ and the $j$-th center $\mathbf{c}$ is defined as:

$$\psi_{\text{Gaussian}}(\mathbf{a}^i, \mathbf{c}^j) = \exp\left(-\frac{(\mathbf{a}^i - \mathbf{c}^j)^2}{2\zeta^2}\right), \tag{3}$$

where $\zeta$ is the width parameter controlling the spread of the Gaussian function. Similarly, the Multiquadric RBF for the $i$-th dimension of $\mathbf{a}$ and the $j$-th center $\mathbf{c}$ is:

$$\psi_{\text{Multiquadric}}(\mathbf{a}^i, \mathbf{c}^j) = \sqrt{(\mathbf{a}^i - \mathbf{c}^j)^2 + \sigma^2}, \tag{4}$$

where $\sigma$ is a constant parameter controlling the shape of the Multiquadric function.

The overall basis function $\phi(\mathbf{a})$ is obtained by summing the weighted RBF values over all dimensions of the treatment and all centers:

$$\phi(\mathbf{a}) = \sum_{i=1}^{d_a} \sum_{j=1}^{d_c} w_{ij} \psi(\mathbf{a}^i, \mathbf{c}^j), \tag{5}$$

where $\psi(\mathbf{a}^i, \mathbf{c}^j)$ can be either the Gaussian RBF or the Multiquadric RBF, and $\{w_{ij}\}$ are the learnable weights associated with each RBF. By incorporating these RBF-based basis functions into the IFC, the model can learn the complex, nonlinear relationships in the temporal dynamics of treatment effects, enabling more accurate estimation of counterfactual outcomes.

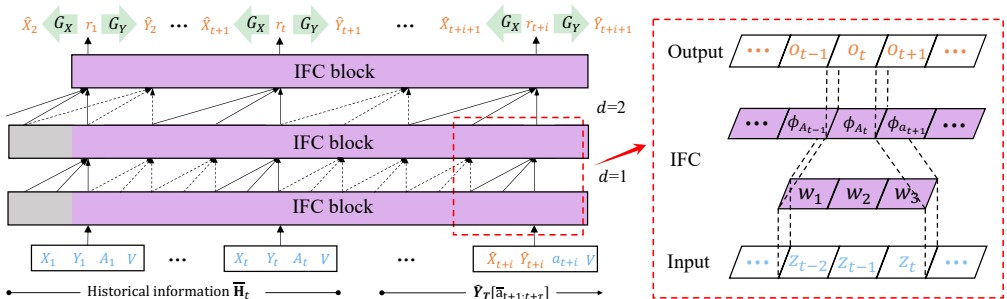

Figure 2: The module in TACIN that learns representations of historical information primarily consists of multiple IFC blocks. To enlarge the receptive field, we employ dilated convolutions, where $d$ denotes the dilation factor. The right side of the figure illustrates the structure of an IFC kernel, which is determined by the treatment nonlinear function $\phi(\mathbf{A}_t)$ at the current time step (abbreviated as $\phi_{\mathbf{A}_t}$ in the figure). Subsequently, counterfactual estimation is performed on the learned representations using feedforward neural networks $G_X$ and $G_Y$.

## 3.2 Addressing Confounding Bias from a Domain Generalization Perspective

Existing methods for causal inference with time-varying treatments often borrow ideas from the static setting and aim to address confounding bias through the lens of domain adaptation. Specifically, these approaches seek to learn representations that align the conditional distributions $P(\Phi(\bar{\mathbf{H}}_t)|\mathbf{A}_t = \mathbf{a}_j)$ across different treatment assignments $\mathbf{a}_j$, thereby removing the association between time-dependent confounders present in the patient history $\bar{\mathbf{H}}_t$ and time-varying treatments $\mathbf{A}_t$ (Bica et al., 2020b; Melnychuk et al., 2022; Wang et al., 2024). However, for this alignment to successfully remove confounding bias, it is assumed that the distribution of the observed histories $\bar{\mathbf{H}}_t$ remains consistent between training and testing (see Remark 1 for a detailed explanation). In the longitudinal setting, this assumption may not hold, limiting the effectiveness of these methods in real-world scenarios. To illustrate this, we can derive the distribution of $\bar{\mathbf{H}}_t$ in the training phase as:

$$P(\bar{\mathbf{H}}_t) = P(\mathbf{X}_1) \prod_{s=1}^{t-1} P_{\text{train}}(\mathbf{A}_s|\bar{\mathbf{H}}_s)P(\mathbf{X}_{s+1}|\bar{\mathbf{H}}_s, \mathbf{A}_s) \tag{6}$$

where $P_{\text{train}}(\mathbf{A}_s|\bar{\mathbf{H}}_s)$ denotes the probability of treatment assignment at step $s$ given the history $\bar{\mathbf{H}}_s$, which is typically determined by doctors following specific treatment policies during training.

However, during the testing phase, given a history $\bar{\mathbf{H}}_{t-\tau}$, we may encounter a new sequence of treatments $\bar{\mathbf{A}}_{t-\tau:t-1}$ of length $\tau$, which results in a trajectory of length $t$. The distribution of this trajectory can be expressed as:

$$P_\tau(\bar{\mathbf{H}}_t) = P(\bar{\mathbf{H}}_{t-\tau}) \prod_{s=t-\tau}^{t-1} P_{\text{test}}(\mathbf{A}_s|\bar{\mathbf{H}}_s)P(\mathbf{X}_{s+1}|\bar{\mathbf{H}}_s, \mathbf{A}_s), \tag{7}$$

where $P_{\text{test}}(\mathbf{A}_s|\bar{\mathbf{H}}_s)$ represents the treatment assignment policy employed during the testing phase (e.g., random assignment), which generally differs from the policy used during training.

The testing phase involves trajectories $\bar{\mathbf{H}}_t$ of length $t$, formed by applying different treatment subsequences of length $\tau$, whose distribution often differs from that observed during training. This discrepancy between the distributions of $\bar{\mathbf{H}}_t$ in the training and testing phases poses a significant challenge for methods that rely on the assumption of distributional consistency, leading to the inability to maintain consistent conditional distributions $P(\Phi(\bar{\mathbf{H}}_t)|\mathbf{A}_t = \mathbf{a}_j)$ across different treatment assignments $\mathbf{a}_j$ in the testing phase.

In contrast to previous work, this paper proposes a novel perspective on addressing confounding bias through the lens of domain generalization. Let $\mathbf{u}_t = [\bar{\mathbf{H}}_t, \mathbf{A}_t] \in \mathcal{U}$, $h : \mathcal{U} \to \mathcal{Y}$ be a hypothesis function for counterfactual estimation, and $l : \mathcal{Y} \times \mathcal{Y} \to \mathbb{R}^+$ be a loss function. We denote the source (observed) domain of $\mathbf{u}_t$ as $D_S$, which follows the factual distribution $P_S(\mathbf{u}_t) = P(\bar{\mathbf{H}}_t|\mathbf{A}_t)P(\mathbf{A}_t)$, and the target domain as $D_T$, which follows the counterfactual distribution $P_T(\mathbf{u}_t) = P_{\text{test}}(\bar{\mathbf{H}}_t|\mathbf{A}_t)P_{\text{test}}(\mathbf{A}_t)$. Let $R_S := \mathbb{E}_{\mathbf{u}_t \sim P_S}[l(h(\mathbf{u}_t), \mathbf{y}_t)]$ denote the factual risk of

a hypothesis $h$, and similarly, let $R_T$ denote the counterfactual risk. Our goal is to find a hypothesis $h$ that minimizes $R_T$.

Previous works (Bica et al., 2020b; Melnychuk et al., 2022; Wang et al., 2024) have often approached this problem from the perspective of domain adaptation. However, based on our earlier analysis, for the target domain $D_T$, we not only lack knowledge of the potential outcomes but also of its distribution, making it infeasible to sample from the observed data. In other words, we are faced with a domain generalization problem, where the model have to generalize to unseen distributions during testing. To address this challenge, we propose a novel approach that constructs a upper bound on the counterfactual risk based on Sharpness-Aware Minimization (SAM) and adversarially generated samples. The specific details are provided in the next section.

### 3.3 GENERALIZATION BOUND ON COUNTERFACTUAL RISK

In order to derive a upper bound on the counterfactual risk $R_T$, we first introduce the concept of SAM (Foret et al., 2021). SAM is a novel optimization technique that seeks to find a flat minimum of the loss function, which has been shown to improve the generalization performance of deep learning models. The key idea behind SAM is to minimize the maximum loss within a neighborhood of the current model parameters, rather than just the loss at the current parameters. Formally, given a model $h_\theta$ parameterized by $\theta$, SAM solves the following optimization problem:

$$\min_\theta \max_{\|\epsilon\| \leq \rho} R_S(h_{\theta+\epsilon}),\tag{8}$$

where $\epsilon$ is a small perturbation to the model parameters, and $\rho$ is a hyperparameter that controls the size of the neighborhood around $\theta$. By solving this minimax optimization problem, SAM finds a set of parameters $\theta^*$ that minimize the maximum loss within a $\rho$-neighborhood of $\theta^*$, resulting in a flatter loss landscape and improved generalization. Additionally, Foret et al. (2021) point out that the factual risk can be bounded as shown in Lemma 1.

**Lemma 1** (Sharpness-Aware Minimization (Foret et al., 2021))**.** *The source risk $R_S(h_\theta)$ is bounded using the following PAC-Bayes generalization bound for any $\rho$ with probability $1 - \delta$:*

$$R_S(h_\theta) \leq \max_{\|\epsilon\| \leq \rho} \hat{R}_S(h_{\theta+\epsilon}) + \gamma(\|\theta\|_2^2/\rho^2),\tag{9}$$

*where $\gamma(\|\theta\|_2^2/\rho^2) = \sqrt{\frac{1}{n-1}\left(k\log\left(1 + \frac{\|\theta\|_2^2}{\rho^2}\left(1 + \sqrt{\frac{\log(n)}{k}}\right)^2\right) + 4\log\frac{n}{\delta} + \tilde{O}(1)\right)}$, and $n$ is the training set size used for calculation of empirical risk $\hat{R}_S(h_\theta)$, $k$ is the number of parameters and $\|\theta\|_2$ is the norm of the weight parameters.*

Given that our primary objective is to optimize the counterfactual error, we first introduce the generalization bounds proposed by Shalit et al. (2017) based on domain adaptation to bridge the gap between the factual and counterfactual risks.

**Lemma 2** (Generalization Bound via IPM (Shalit et al., 2017))**.** *Let $\Phi : \mathcal{U} \to \mathcal{R}$ be a one-to-one representation function, with inverse $\Psi$. Let $h : \mathcal{U} \to \mathcal{Y}$ and $f : \mathcal{R} \to \mathcal{Y}$ be hypothesis and $H, F$ be the sets of all possible hypothesis (i.e. Hypothesis Space) over $\mathcal{U}$ and $\mathcal{R}$ respectively. Let $\mathcal{G}$ be a family of functions $g : \mathcal{R} \to \mathcal{Y}$. Define the Integral Probability Metric(IPM) of two distributions:*

$$\text{IPM}_G(P_S^\Phi, P_T^\Phi) = \sup_{g \in G} \left| \mathbb{E}_{\mathbf{r} \sim P_S^\Phi}[g(\mathbf{r})] - \mathbb{E}_{\mathbf{r} \sim P_T^\Phi}[g(\mathbf{r})] \right|,\tag{10}$$

*where $\mathbf{r} = \Phi(\mathbf{u})$. Suppose there exists a constant $B_\Phi > 0$, such that $\forall h \in H, y \in \mathcal{Y}, \frac{1}{B_\Phi}l(f(\mathbf{r}), y) \in G$. Then we have:*

$$R_T(h) \leq R_S(h) + B_\Phi \cdot \text{IPM}_G(P_S^\Phi, P_T^\Phi).\tag{11}$$

However, as analyzed in Section 3.2, we cannot sample from the distribution $P_T^\Phi$, rendering the bound provided in Equation 11 impractical for direct application. To address this issue, we propose to generate adversarial examples using the Fast Gradient Sign Method (FGSM) (Goodfellow et al., 2015) and provide the following lemma to upper bound $\text{IPM}_G(P_S^\Phi, P_T^\Phi)$.

**Lemma 3** (Generalization Bound via Adversarial Distribution). *Let $P_S^\Phi$ and $P_T^\Phi$ be the source and target distributions of the representation $\mathbf{r}$, respectively. Consider the following conditions:*

1. *For all $f \in F$ and $y \in \mathcal{Y}$, the loss function $l(f(\mathbf{r}), y) \in G$;*

2. *$G$ is a set of Lipschitz functions, i.e., there exists $L > 0$ such that for all $g \in G$ and $\mathbf{r}_1, \mathbf{r}_2 \in \mathcal{R}$, $|g(\mathbf{r}_1) - g(\mathbf{r}_2)| \leq L|\mathbf{r}_1 - \mathbf{r}_2|$;*

3. *$\mathbb{E}_{\mathbf{r} \sim P_A^\Phi}|\mathbf{r}| + \mathbb{E}_{\mathbf{r} \sim P_S^\Phi}|\mathbf{r}| < \infty$;*

4. *For all $f \in F$, $\max_y \left| \mathbb{E}_{\mathbf{r} \sim P_A^\Phi}[l(f(\mathbf{r}), y)] - \mathbb{E}_{\mathbf{r} \sim P_S^\Phi}[l(f(\mathbf{r}), y)] \right| > 0$.*

*Let the adversarial samples be generated by $\mathbf{r}_{adv} = \mathbf{r} + \epsilon \cdot sign(\nabla_\mathbf{r} l(f(\mathbf{r}), y))$ and $\mathbf{r}_{adv} \sim P_A^\Phi$, where $\epsilon = \arg\max_\epsilon [l(f(\mathbf{r}_{adv}), y)) - l(f(\mathbf{r}), y))]$. If conditions (1)-(4) hold, then there exists a constant $M_\Phi > 0$ such that*

$$\text{IPM}_G(P_S^\Phi, P_T^\Phi) \leq M_\Phi \cdot \text{IPM}_G(P_S^\Phi, P_A^\Phi). \tag{12}$$

Lemma 3 indicates that under certain conditions, we can approximate $\text{IPM}_G(P_S^\Phi, P_T^\Phi)$ using $\text{IPM}_G(P_S^\Phi, P_A^\Phi)$, where $P_A^\Phi$ denotes the distribution of adversarial examples generated from the source distribution $P_S^\Phi$. Among these conditions, the third one ensures that the adversarial and target distributions are *absolutely integrable*, a property satisfied by probability distributions with finite variance. This condition guarantees that the expected values of functions under these distributions are well-defined and finite. Furthermore, the fourth condition is reasonable because adversarial examples are generated to maximize the model's loss, implying that the loss incurred by the adversarial distribution is likely to be strictly greater than that of the source distribution. This condition ensures that the adversarial distribution provides a meaningful upper bound on the target risk.

Under these conditions, Lemma 3 provides a tractable way to approximate the distributional discrepancy between the source and target domains using the IPM between the source distribution and its adversarially perturbed counterpart. This approximation allows us to optimize the representation $\Phi$ to minimize the distributional discrepancy, even without having access to samples from the target domain. To further improve the model's generalization ability on unseen domains, we derive a upper bound on the factual risk by combining the results from Lemma 1, Lemma 2, and Lemma 3.

**Theorem 1** (Generalization Bound via SAM and Adversarial Distribution). *Under the conditions of Lemma 2 and Lemma 3, we have:*

$$R_T(h_\theta) \leq \max_{\|\epsilon\| \leq \rho} \hat{R}_S(h_{\theta+\epsilon}) + \alpha_\Phi \text{IPM}(P_S^\Phi, P_A^\Phi) + \gamma(\|\theta\|_2^2/\rho^2), \tag{13}$$

*where $\alpha_\Phi > 0$ is a constant.*

In the following, we introduce how to leverage the IFC kernel to construct models for counterfactual estimation over time, and optimize them based on theoretical results.

### 3.4 COUNTERFACTUAL ESTIMATION

As shown in Figure 2, we propose the Temporal Adaptive Convolutional Intervention Network (TACIN). Following the methodology of Bai et al. (2018), TACIN consists of a representation module $\Phi(\cdot)$, which employs residual connections (He et al., 2016) to concatenate multiple IFC blocks:

$$\mathbf{o} = \text{Activation}(B(\mathbf{z}) + F(\mathbf{z})). \tag{14}$$

When input $\mathbf{z}$ and output $F(\mathbf{z})$ dimensions match, $B$ is an identity map. Otherwise, $B$ becomes a $1 \times 1$ convolution to align dimensions.

TACIN employs an autoregressive recursive strategy (Chevillon, 2007; Taieb & Atiya, 2015) for multi-step prediction, which has also been adopted by (Li et al., 2021; Wang et al., 2024). During the training process, we need to predict the output and time-varying covariates at the next time step. TACIN defines a feedforward neural network $G_Y$ to decode the predicted output from the representation $\Phi(\mathbf{u}_t)$. For notational simplicity, we denote $\Phi(\mathbf{u}_t)$ as $\mathbf{r}_t$. We use the Mean Squared Error (MSE) to define the following loss function:

$$\mathcal{L}_Y^t(\theta) = \|\mathbf{Y}_{t+1} - G_Y(\mathbf{r}_t)\|^2. \tag{15}$$

TACIN employs two feedforward neural networks, $G_X$ and $J_X$, for covariate prediction. Specifically, $G_X$ is responsible for decoding the expected covariates from $\mathbf{r}_t$. Drawing inspiration from Wang et al. (2024), we utilize $J_X$ to design a smoothing mechanism. This mechanism, influenced by the gating mechanism in GRUs (Cho et al., 2014), aims to adapt to the varying trends of different covariates. Analogously, we define the following loss function:

$$\mathcal{L}_X^t(\theta) = \|\mathbf{X}_{t+1} - (\eta G_X(\mathbf{r}_t) + (1 - \eta)\mathbf{X}_t)\|^2, \tag{16}$$

where $\eta = \text{Sigmoid}(J_X(\mathbf{r}_t))$ is the smoothing factor. Based on Theorem 1, we propose the following objective function:

$$\mathcal{L} = \frac{1}{N} \sum_{i \in \mathcal{D}} \sum_{t=1}^{T} \max_{\|\epsilon\| \le \rho}(\mathcal{L}_Y^{t(i)}(\theta + \epsilon) + \lambda_1 \mathcal{L}_X^{t(i)}(\theta + \epsilon)) + \lambda_2 \sum_{t=1}^{T} \text{IPM}(\mathbf{r}_t^{i \in \mathcal{D}}, \mathbf{r}_{t_{adv}}^{i \in \mathcal{D}}) + \lambda_3 \mathscr{R}(\theta), \tag{17}$$

where $\mathscr{R}(\theta)$ is the regularization term. We implement TACIN using the Pytorch Lightning framework and employ the Adam algorithm (Kingma & Ba, 2014) for gradient optimization. Upon completion of the training phase, TACIN generates one-step-ahead predictions and employs an autoregressive strategy for multi-step-ahead predictions.

## 4 EXPERIMENTS

In this section, we validate the effectiveness of TACIN through comparative experiments on simulated datasets. Subsequently, we conduct ablation studies to examine the efficacy of our proposed theory, specifically the role of Equation 17 in mitigating confounding bias.

**Baselines.** We compare our method against state-of-the-art models for counterfactual estimation over time: **RMSN** (Lim et al., 2018), **CRN** (Bica et al., 2020b), **G-Net** (Li et al., 2021), **CT** (Melnychuk et al., 2022), and **ACTIN** (Wang et al., 2024). To ensure fair comparison, we perform hyperparameter tuning for all baselines (see details in Appendix E).

### 4.1 EXPERIMENTS WITH FS-TUMOR DATA

**Data.** The fully-synthetic tumor (FS-Tumor) growth dataset, constructed using a pharmacokinetic-pharmacodynamic model (Geng et al., 2017), provides a realistic simulation of the combined effects of chemotherapy and radiotherapy on lung cancer patients. This dataset has gained significant attention in the research community and has been extensively utilized for evaluating various causal inference methods in studies such as (Lim et al., 2018; Bica et al., 2020b; Melnychuk et al., 2022; Wang et al., 2024). One notable feature of this advanced bio-mathematical model is the inclusion of a parameter $\gamma$, which allows for the control of time-varying confounding in the dataset. By adjusting the value of $\gamma$, researchers can simulate scenarios where historical data have varying degrees of influence on treatment allocation. As $\gamma$ increases, the confounding bias becomes more pronounced, as past information plays a more dominant role in determining the course of treatment.

In the original dataset, both radiotherapy and chemotherapy interventions are binary. However, in real-world clinical settings, these treatments often involve varying dosages. To better reflect this complexity, we have adapted the interventions in our study to be continuous rather than binary. This modification allows for a more nuanced representation of treatment intensities, aligning our simulations more closely with clinical realities. For a detailed description of the dataset generation process, including this adaptation, please refer to Appendix D.1.

**Results.** Figure 3 illustrates the performance comparison between TACIN and the baseline models on the FS-Tumor dataset. The results demonstrate that TACIN exhibits superior or highly competitive performance across different levels of time-varying confounding ($\gamma = 0$, 2, and 4) and prediction steps (1 to 6 steps). In most cases, TACIN and the baseline models share a similar performance trend, typically reaching a peak RMSE around the 3rd step and then showing a decreasing trend in long-term predictions. It is noteworthy that this trend does not stem from a reduction in the difficulty of long-term prediction tasks, but rather may be attributed to the reduction of tumor size after the application of treatment interventions, leading to a decrease in RMSE.

It is especially important to note that as confounding bias becomes more severe, treatment allocation grows increasingly imbalanced, complicating the capture of temporal treatment interactions. In such

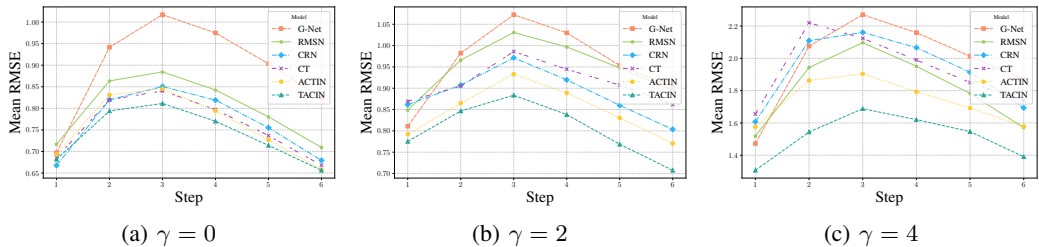

(a) $\gamma = 0$        (b) $\gamma = 2$        (c) $\gamma = 4$

Figure 3: Performance comparison of the TACIN model with alternative models for 1-step to 6-step predictions on FS-Tumor datasets. Results are presented for three levels of time-varying confounding factor: $\gamma = 0$, $\gamma = 2$, and $\gamma = 4$. Each subplot shows the mean RMSE over ten runs for a specific $\gamma$ value, with the x-axis representing the prediction steps (1 to 6) and the y-axis showing the RMSE.

Table 1: $\tau$-step-ahead prediction results for experiments with CISD dataset. Shown: RMSE as mean $\pm$ standard deviation over ten runs.

| | $\tau = 1$ | $\tau = 2$ | $\tau = 3$ | $\tau = 4$ | $\tau = 5$ | $\tau = 6$ |
|---|---|---|---|---|---|---|
| RMSN | $1.85 \pm 0.56$ | $3.03 \pm 0.76$ | $3.50 \pm 0.83$ | $3.86 \pm 0.79$ | $4.59 \pm 1.44$ | $5.21 \pm 1.38$ |
| CRN | $1.67 \pm 0.39$ | $2.56 \pm 0.78$ | $3.07 \pm 0.90$ | $3.52 \pm 0.89$ | $4.19 \pm 1.53$ | $4.77 \pm 1.48$ |
| G-Net | $2.06 \pm 0.54$ | $2.92 \pm 0.84$ | $3.48 \pm 1.03$ | $3.80 \pm 1.01$ | $4.42 \pm 1.48$ | $5.11 \pm 1.52$ |
| CT | $1.85 \pm 0.49$ | $2.81 \pm 0.83$ | $3.30 \pm 0.87$ | $3.63 \pm 0.87$ | $4.20 \pm 1.35$ | $4.85 \pm 1.37$ |
| ACTIN | $0.89 \pm 0.31$ | $1.48 \pm 0.61$ | $1.94 \pm 0.69$ | $2.58 \pm 0.84$ | $3.31 \pm 1.22$ | $4.24 \pm 1.83$ |
| TACIN | $\mathbf{0.71 \pm 0.14}$ | $\mathbf{0.97 \pm 0.28}$ | $\mathbf{1.31 \pm 0.47}$ | $\mathbf{1.66 \pm 0.49}$ | $\mathbf{2.03 \pm 0.62}$ | $\mathbf{2.32 \pm 0.66}$ |

scenarios, the benefits of TACIN are particularly pronounced, clearly demonstrating the superiority of its distinctive IFC approach. For more comprehensive experimental results on the FS-Tumor dataset, please refer to Appendix F.

## 4.2 EXPERIMENTS WITH CISD DATA

**Data.** The Continuous Intervention Synthetic Dataset (CISD) (Wang et al., 2024) is a synthetic time series dataset designed to simulate the effects of continuous interventions. In this dataset, the generation of treatment variables relies on nonlinear transformations of historical covariate data and the addition of noise, reflecting confounding bias through a Beta distribution. Furthermore, the generation of covariate and outcome variables significantly depends on complex nonlinear transformations that not only capture the dynamic influence of historical covariates and treatment data but also enhance the realism of the dataset. These nonlinear processing techniques make the CISD dataset a useful tool for investigating the intricate dynamic effects of continuous interventions, making it particularly suitable for simulating treatment effects in real-world scenarios. For a detailed description of the dataset generation process, please refer to Appendix D.2.

**Results.** Table 1 presents the $\tau$-step prediction results on the CISD dataset. Compared to the FS-Tumor dataset, CISD has a higher dimensionality of covariates, resulting in more complex temporal treatment interactions. As the number of prediction steps increases, this complexity leads to greater challenges in making accurate predictions, which can be observed from the performance of different models. The results demonstrate that TACIN consistently outperforms other baseline models across all prediction steps. Notably, as the number of prediction steps increases, the advantage of TACIN over other baseline models becomes more prominent. This highlights the effectiveness of TACIN in capturing temporal treatment interactions by emphasizing the role of treatments through the IFC approach, thereby significantly improving model performance. In contrast, other models such as ACTIN, although attempting to address these interactions, still treat treatments as general inputs, limiting their potential to handle nonlinear complexities.

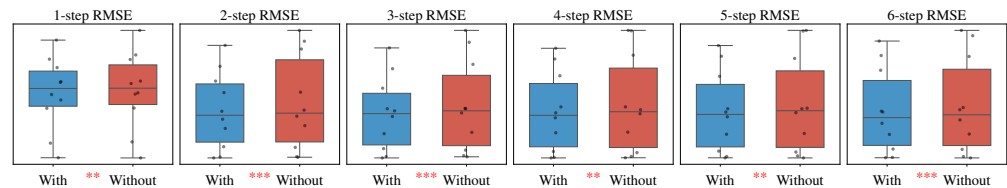

Figure 4: The ablation study results for TACIN on the FS-Tumor dataset with $\gamma = 4$.

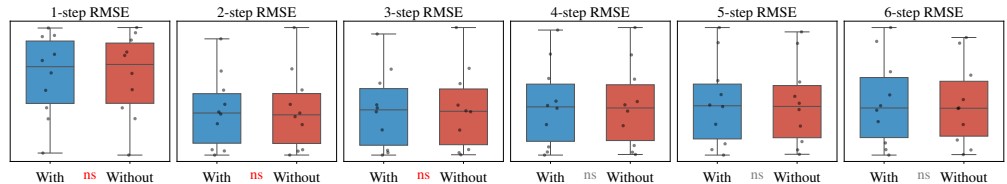

Figure 5: The ablation study results for ACTIN on the FS-Tumor dataset with $\gamma = 4$.

### 4.3 ABLATION STUDIES

To further investigate the effectiveness of our proposed theory, we conducted an ablation study on the FS-Tumor dataset with $\gamma = 4$. We compared the performance of the TACIN model with and without the objective function in Equation 17 to evaluate the impact of the proposed theory on alleviating confounding bias. To contrast with balancing strategies from the perspective of domain adaptation, we also selected ACTIN as a representative for the ablation experiment. The results are shown in Figure 4 and Figure 5.

The "with" and "without" represent the presence and absence of the objective function in TACIN or the balancing strategy in ACTIN, respectively. We conducted the Wilcoxon signed-rank test on the results of ten runs (datasets with different seeds) to determine whether the "with" configuration significantly outperforms the "without" configuration. The symbols *, **, and *** denote $p$-values less than 0.1, 0.05, and 0.01, respectively, indicating varying levels of statistical significance. The "ns" label signifies non-significant results. Cases with lower average RMSE under the "with" configuration are highlighted in red.

The results in Figure 4 show that TACIN with our proposed objective function consistently achieves superior performance when $\tau$ ranges from 1 to 6, and the results are statistically significant. Examining the distribution of experimental results reveals that our objective function leads to stable improvements across datasets generated with different random seeds, especially on those more challenging datasets (higher RMSE). In contrast, Figure 5 demonstrates that the balancing strategie of ACTIN slightly outperforms the baseline when $\tau$ is between 1 and 3, but the difference is not statistically significant. The distribution indicate that the impact of the balancing strategie on model performance is not evident, particularly when $\tau$ is large. This observation aligns with our theoretical analysis, suggesting that approaching the problem from the perspective of domain generalization, rather than domain adaptation, is more effective in reducing time-varying confounding bias.

## 5 CONCLUSION

In this paper, to overcome the challenges posed by confounding bias and temporal complexities in estimating counterfactuals over time, we propose a novel model called TACIN. By encoding treatment information within the IFC kernel, TACIN can better emphasize the role of treatments, effectively capturing the complexities caused by temporal treatment interactions. Furthermore, TACIN provides a new perspective on mitigating time-varying confounding bias by bounding the counterfactual estimation error from the viewpoint of domain generalization. We demonstrate the effectiveness of TACIN from both theoretical and practical aspects, highlighting its potential for accurate counterfactual estimation in longitudinal settings.

## REPRODUCIBILITY STATEMENT

All theoretical proofs in this paper can be found in Appendix C. To ensure the reproducibility of the experimental results, we provide all the code used to reproduce the experimental results in an anonymous repository at `https://anonymous.4open.science/r/TACIN-2D20`. Readers can follow the detailed steps provided in the repository to reproduce the experimental results presented in this paper.

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

# A RELATED WORK

## A.1 CAUSAL INFERENCE WITH LONGITUDINAL DATA

Recently, significant progress has been made in the field of longitudinal data analysis by incorporating neural network techniques for counterfactual estimation. For instance, RMSN (Lim et al., 2018) combines two propensity networks and employs an inverse probability weighting (IPTW) based training method in its prediction model. G-Net (Li et al., 2021) enhances the traditional G-computation technique through a deep learning framework. Similarly, CRN (Bica et al., 2020b), which is also based on RNN networks, focuses on learning balanced representations to mitigate confounding bias. In contrast to these works, CT (Melnychuk et al., 2022) proposes utilizing a more powerful transformer model to better handle long-term dependencies, while ACTIN (Wang et al., 2024) introduces a dual-module framework that effectively enhances the ability of simple models to handle complex temporal treatment interactions. It is worth noting that CRN, CT, and ACTIN all adopt the idea of domain adaptation to learn balanced representations by eliminating the association between historical information and current treatment assignments to alleviate confounding bias.

There have been significant advances in causal inference based on longitudinal data, albeit with different settings from ours (Bica et al., 2020a; Hatt & Feuerriegel, 2024; Frauen et al., 2023; Seedat et al., 2022; Meng et al., 2023; Hess et al., 2024; Chen et al., 2023). These works primarily focus on addressing time-varying confounding factors in longitudinal observational data to achieve accurate causal effect estimation. For instance, the Time Series Deconfounder (Bica et al., 2020a) and Deep-ACE (Frauen et al., 2023) utilize recurrent neural networks and end-to-end deep learning models, respectively, combining latent variable inference and iterative G-computation formula to adjust for the impact of time-varying confounders. For irregularly sampled longitudinal data, Treatment Effect Neural Controlled Differential Equation (TE-CDE) (Seedat et al., 2022) and Bayesian Neural Controlled Differential Equation (BNCDE) (Hess et al., 2024) propose differential equation-based modeling approaches to estimate counterfactual outcomes, with BNCDE additionally providing uncertainty estimates of treatment effects using Bayesian uncertainty quantification. Chen et al. (2023) introduces a multi-task Gaussian process model that captures relationships between treatments, subjects, and temporal variations by defining multi-task priors, thereby estimating dynamic treatment effects. The aforementioned works mainly focus on causal inference under standard conditions, while COSTAR (Meng et al., 2023) considers the problem of counterfactual estimation under distributional shifts and improves model performance by introducing self-supervised learning, which sets it apart from other studies. Overall, these research efforts enrich the methods for causal inference on longitudinal data, providing new perspectives for addressing issues such as time-varying confounders, irregularly sampled data, and distributional shifts.

## A.2 CAUSAL INFERENCE WITH STAIC DATA

In static settings, a key challenge is that treatments are often assigned based on unit-specific covariates, leading to imbalanced covariate distributions across treatment groups. Addressing this imbalance is crucial for ensuring the reliability of causal inference (Johansson et al., 2016; Shalit et al., 2017; Yoon et al., 2018). Inspired by domain adaptation, a line of research tackles this challenge by learning balanced representations between treatment and control groups. For instance, Shalit et al. (Shalit et al., 2017) learn balanced representations by minimizing the Integral Probability Metric (IPM) distance between the distributions of treated and control groups. These representation learning-based methods aim to eliminate distributional differences between treatment groups, thereby mitigating the impact of confounding bias on causal effect estimation.

Another challenge is how to differentiate treatment variables from other covariates within neural network architectures, which has been noted in previous works (Shalit et al., 2017; Schwab et al., 2020; Nie et al., 2020). For binary treatments, Shalit et al. (Shalit et al., 2017) introduce the Counterfactual Regression (CFR) framework, which learns a shared representation followed by two separate "heads" to predict post-treatment and control outcomes, effectively mitigating the potential loss of treatment information in high-dimensional latent representations. This approach has been widely adopted in subsequent studies (Louizos et al., 2017; Shi et al., 2019; Hassanpour & Greiner, 2019). For continuous treatments, Nie et al. (Nie et al., 2020) propose the Varying Coefficient Network (VCNet), allowing the prediction head weights to be continuous functions of the treatment, simi-

larly emphasizing the impact of treatment information. However, in longitudinal settings, how to design models specifically for treatment information remains an open question.

### A.3 DOMAIN GENERALIZATION

Domain generalization (DG) aims to learn a model that can perform well on unseen target domains, which is crucial for addressing distribution shift problems in real-world scenarios (Blanchard et al., 2011; Muandet et al., 2013). Many methods have been proposed to tackle this issue, including learning domain-invariant representations (Motiian et al., 2017; Matsuura & Harada, 2020), data augmentation-based approaches (Xu et al., 2020; Volpi et al., 2018), and robustness training-based methods (Shi et al., 2021; Foret et al., 2021; Zhang et al., 2024). Despite the significant progress made by these methods in tasks such as image classification, their application to time series data remains challenging. Compared to static data, the dynamic generative process of time series data makes their distribution more complex, which may be the reason why the strategy of learning domain-invariant representations is less effective in mitigating time-varying confounding bias. The flatness-related methods, such as Sharpness-Aware Minimization (SAM) (Foret et al., 2021), adopted in this paper aim to address the effects of domain shifts by identifying flat minima, seeking regions in the loss landscape where small perturbations in the input have minimal impact on the model's predictions. By leveraging flat minima, these methods enhance the model's robustness to domain variations. It is worth noting that this approach is just one possible solution path, and from the perspective of domain generalization, it can bring us more possibilities for mitigating time-varying confounding bias.

## B ASSUMPTIONS

To reliably identify treatment effects from observational data, it is essential to adopt the following assumptions, as delineated in related literature (Lim et al., 2018; Bica et al., 2020b; Li et al., 2021; Melnychuk et al., 2022; Wang et al., 2024):

**Assumption A.1** (**Consistency**). *At time $t + 1$, the observed outcome $\mathbf{Y}_{t+1}$ is identical to the potential outcome $\mathbf{Y}_{t+1}[\mathbf{a}_t]$ under the treatment $\mathbf{a}_t$ at time $t$, i.e., $\mathbf{Y}_{t+1} = \mathbf{Y}_{t+1}[\mathbf{a}_t]$.*

**Assumption A.2** (**Sequential Overlap**). *The probability of receiving any treatment $\mathbf{a}_t$ at time $t$ is always positive, i.e., $0 < P(\mathbf{A}_t = \mathbf{a}_t \mid \bar{\mathbf{H}}_t = \bar{\mathbf{h}}_t) < 1$, $\forall \mathbf{a}_t \in \mathcal{A}$ if $P(\bar{\mathbf{H}}_t = \bar{\mathbf{h}}_t) > 0$, where $\bar{\mathbf{h}}_t$ is a realization of $\bar{\mathbf{H}}_t$.*

**Assumption A.3** (**Sequential Ignorability**). *The treatment at any time $t$ is independent of the potential outcomes at time $t + 1$, given the observed history, i.e., $\mathbf{A}_t \perp \mathbf{Y}_{t+1}[\mathbf{a}_t] \mid \bar{\mathbf{H}}_t$, $\forall \mathbf{a}_t \in \mathcal{A}$. This indicates the absence of unobserved confounders that affect both treatment and outcome.*

## C PROOFS

**Lemma A.1** (Sharpness Aware Minimization (Foret et al., 2021)). *The source risk $R_S(h_\theta)$ is bounded using the following PAC-Bayes generalization bound for any $\rho$ with probability $1 - \delta$:*

$$R_S(h_\theta) \leq \max_{\|\epsilon\| \leq \rho} \hat{R}_S(h_{\theta+\epsilon}) + \gamma(\|\theta\|_2^2/\rho^2), \tag{18}$$

*where $\gamma(\|\theta\|_2^2/\rho^2) = \sqrt{\frac{1}{n-1}\left(k\log\left(1 + \frac{\|\theta\|_2^2}{\rho^2}\left(1 + \sqrt{\frac{\log(n)}{k}}\right)^2\right) + 4\log\frac{n}{\delta} + \tilde{O}(1)\right)}$, and $n$ is the training set size used for calculation of empirical risk $\hat{R}_S(h_\theta)$, $k$ is the number of parameters and $\|\theta\|_2$ is the norm of the weight parameters.*

*Proof.* See Theorem 2 in the paper sharpness aware minimization (Foret et al., 2021). □

**Lemma A.2** (Generalization Bound via IPM (Shalit et al., 2017)). *Let $\Phi : \mathcal{U} \rightarrow \mathcal{R}$ be a one-to-one representation function, with inverse $\Psi$. Let $h : \mathcal{U} \rightarrow \mathcal{Y}$ and $f : \mathcal{R} \rightarrow \mathcal{Y}$ be hypothesis and $H, F$*

*be the sets of all possible hypothesis (i.e. Hypothesis Space) over $\mathcal{U}$ and $\mathcal{R}$ respectively. Let $\mathcal{G}$ be a family of functions $g : \mathcal{R} \to \mathcal{Y}$. Define the Integral Probability Metric(IPM) of two distributions:*

$$\text{IPM}_G(P_S^\Phi, P_T^\Phi) = \sup_{g \in G} \left| \mathbb{E}_{\mathbf{r} \sim P_S^\Phi}[g(\mathbf{r})] - \mathbb{E}_{\mathbf{r} \sim P_T^\Phi}[g(\mathbf{r})] \right|, \tag{19}$$

*where $\mathbf{r} = \Phi(\mathbf{u})$. Suppose there exists a constant $B_\Phi > 0$, such that $\forall h \in H, y \in \mathcal{Y}, \frac{1}{B_\Phi} l(f(\mathbf{r}), y) \in G$. Then we have:*

$$R_T(h) \le R_S(h) + B_\Phi \cdot \text{IPM}_G(P_S^\Phi, P_T^\Phi). \tag{20}$$

*Proof.* Since $\Phi(\cdot)$ is a one-to-one function and $\mathbf{r} = \Phi(\mathbf{u}), \mathbf{u} = \Psi(\mathbf{r})$,

$$\mathbb{E}_{\mathbf{u} \sim P_S}[\ell(h(\mathbf{u}), y)] = \mathbb{E}_{\mathbf{r} \sim P_S^\Phi}[\ell(f(\mathbf{r}), y)],$$

$$\mathbb{E}_{\mathbf{u} \sim P_T}[\ell(h(\mathbf{u}), y)] = \mathbb{E}_{\mathbf{r} \sim P_T^\Phi}[\ell(f(\mathbf{r}), y]. \tag{21}$$

Then $\forall h \in H$,

$$R_T(h) - R_S(h) = \mathbb{E}_{\mathbf{u} \sim P_T}[\ell(h(\mathbf{u}), y)] - \mathbb{E}_{\mathbf{u} \sim P_S}[\ell(h(\mathbf{u}), y)] \tag{22}$$

$$= \mathbb{E}_{\mathbf{r} \sim P_T^\Phi}[\ell(f(\mathbf{r}), y)] - \mathbb{E}_{\mathbf{r} \sim P_S^\Phi}[\ell(f(\mathbf{r}), y)] \tag{23}$$

$$\le |\mathbb{E}_{\mathbf{r} \sim P_T^\Phi}[\ell(f(\mathbf{r}), y)] - \mathbb{E}_{\mathbf{r} \sim P_S^\Phi}[\ell(f(\mathbf{r}), y)]| \tag{24}$$

$$\le B_\Phi \sup_{g \in G} \left| \mathbb{E}_{\mathbf{r} \sim P_S^\Phi}[g(\mathbf{r})] - \mathbb{E}_{\mathbf{r} \sim P_T^\Phi}[g(\mathbf{r})] \right| \tag{25}$$

$$= B_\Phi \cdot \text{IPM}_G(P_S^\Phi, P_T^\Phi). \tag{26}$$

$\square$

**Lemma A.3** (Generalization Bound via Adversarial Distribution). *Let the adversarial samples are generated by $\mathbf{r}_{adv} = \mathbf{r} + \epsilon \cdot sign(\nabla_\mathbf{r} l(f(\mathbf{r}), y))$ and $\mathbf{r}_{adv} \sim P_A^\Phi$, where $\epsilon = \arg\max_\epsilon [l(f(\mathbf{r}_{adv}), y)) - l(f(\mathbf{r}), y))]$. Suppose (1) $\forall f \in F, y \in \mathcal{Y}$, the loss function $l(f(\mathbf{r}), y) \in G$, (2) $G$ is a set of Lipschitz functions, i.e., $\exists L > 0$, s.t. $\forall g \in G, \mathbf{r}_1, \mathbf{r}_2 \in \mathcal{R}, |g(\mathbf{r}_1) - g(\mathbf{r}_2)| \le L|\mathbf{r}_1 - \mathbf{r}_2|$,(3) $\mathbb{E}_{\mathbf{r} \sim P_A^\Phi}|\mathbf{r}| + \mathbb{E}_{\mathbf{r} \sim P_S^\Phi}|\mathbf{r}| < \infty$, (4) $\forall f \in F, \max_y \left| \mathbb{E}_{\mathbf{r} \sim P_A^\Phi}[l(f(\mathbf{r}), y)] - \mathbb{E}_{\mathbf{r} \sim P_S^\Phi}[l(f(\mathbf{r}), y)] \right| > 0$. Then, there exists a constant $M_\Phi > 0$, such that*

$$\text{IPM}_G(P_S^\Phi, P_T^\Phi) \le M_\Phi \cdot \text{IPM}_G(P_S^\Phi, P_A^\Phi). \tag{27}$$

*Proof.* It suffices to prove:

$\forall g \in G, \exists M_\Phi > 0$ and $g^* \in G$, such that

$$\left| \mathbb{E}_{\mathbf{r} \sim P_S^\Phi}[g(\mathbf{r})] - \mathbb{E}_{\mathbf{r} \sim P_T^\Phi}[g(\mathbf{r})] \right| \le M_\Phi \left| \mathbb{E}_{\mathbf{r} \sim P_S^\Phi}[g^*(\mathbf{r})] - \mathbb{E}_{\mathbf{r} \sim P_A^\Phi}[g^*(\mathbf{r})] \right|. \tag{28}$$

In fact,

$$\text{L.H.S} = \left| \mathbb{E}_{\mathbf{r} \sim P_S^\Phi}[g(\mathbf{r})] - \mathbb{E}_{\mathbf{r} \sim P_T^\Phi}[g(\mathbf{r})] \right| \tag{29}$$

$$= \left| \mathbb{E}_{\mathbf{r} \sim P_S^\Phi}[g(\mathbf{r})] - \mathbb{E}_{\mathbf{r} \sim P_S^\Phi}[g(\mathbf{r}_0)] + \mathbb{E}_{\mathbf{r} \sim P_T^\Phi}[g(\mathbf{r}_0)] - \mathbb{E}_{\mathbf{r} \sim P_T^\Phi}[g(\mathbf{r})] \right| \tag{30}$$

$$\le \left| \mathbb{E}_{\mathbf{r} \sim P_S^\Phi}[g(\mathbf{r})] - \mathbb{E}_{\mathbf{r} \sim P_S^\Phi}[g(\mathbf{r}_0)] \right| + \left| \mathbb{E}_{\mathbf{r} \sim P_T^\Phi}[g(\mathbf{r}_0)] - \mathbb{E}_{\mathbf{r} \sim P_T^\Phi}[g(\mathbf{r})] \right| \tag{31}$$

$$\le \mathbb{E}_{\mathbf{r} \sim P_S^\Phi}|g(\mathbf{r}) - g(\mathbf{r}_0)| + \mathbb{E}_{\mathbf{r} \sim P_T^\Phi}|g(\mathbf{r}) - g(\mathbf{r}_0)| \tag{32}$$

$$\le L(\mathbb{E}_{\mathbf{r} \sim P_S^\Phi}|\mathbf{r} - \mathbf{r}_0| + \mathbb{E}_{\mathbf{r} \sim P_T^\Phi}|\mathbf{r} - \mathbf{r}_0|) \tag{33}$$

$$\le 2L(\mathbb{E}_{\mathbf{r} \sim P_S^\Phi}|\mathbf{r}| + \mathbb{E}_{\mathbf{r} \sim P_T^\Phi}|\mathbf{r}|). \tag{34}$$

Let $M_{AS} = \max_y \left| \mathbb{E}_{\mathbf{r} \sim P_A^\Phi}[\ell(f(\mathbf{r}), y)] - \mathbb{E}_{\mathbf{r} \sim P_S^\Phi}[\ell(f(\mathbf{r}), y)] \right|,$

$M_{TS} = 2L(\mathbb{E}_{\mathbf{r} \sim P_T^\Phi}|\mathbf{r}| + \mathbb{E}_{\mathbf{r} \sim P_S^\Phi}|\mathbf{r}|)$

By condition (3) and (4), $M_{TS} < \infty, M_{AS} > 0$.

Taking $M_\Phi = M_{TS}/M_{AS}$, we have $M > 0$, and

$$\text{L.H.S} \leq M_\Phi \cdot M_{AS}. \tag{35}$$

Since $\forall f \in F, y \in \mathcal{Y}$, the loss function $\ell(f(\mathbf{r}), y) \in G$,

$$\left| \mathbb{E}_{\mathbf{r} \sim P_A^\Phi}[\ell(f(\mathbf{r}), y)] - \mathbb{E}_{\mathbf{r} \sim P_S^\Phi}[\ell(f(\mathbf{r}), y)] \right| \leq \sup_{g \in G} \left| \mathbb{E}_{\mathbf{r} \sim P_A^\Phi}[g(\mathbf{r})] - \mathbb{E}_{\mathbf{r} \sim P_S^\Phi}[g(\mathbf{r})] \right| = \text{IPM}_G(P_A^\Phi, P_S^\Phi) \tag{36}$$

Therefore,

$$M_{AS} \leq \text{IPM}_G(P_A^\Phi, P_S^\Phi). \tag{37}$$

Thus, L.H.S $\leq$ R.H.S. $\qquad \square$

**Theorem A.4** (Generalization Bound via SAM and Adversarial Distribution). *Under the conditions of Lemma A.2 and Lemma A.3, we have:*

$$R_T(h_\theta) \leq \max_{\|\epsilon\| \leq \rho} \hat{R}_S(h_{\theta+\epsilon}) + \alpha_\Phi \text{IPM}(P_S^\Phi, P_A^\Phi) + \gamma(\|\theta\|_2^2/\rho^2), \tag{38}$$

*where $\alpha_\Phi > 0$ is a constant.*

*Proof.* From the generalization bound for domain adaptation we have the following result according to Lemma A.2:

$$R_T(h) \leq R_S(h) + B_\Phi \cdot \text{IPM}_G(P_S^\Phi, P_T^\Phi). \tag{39}$$

The first term of R.H.S of Equation 39 can be bounded as stated in Lemma A.1,

$$R_S(h_\theta) \leq \max_{\|\epsilon\| \leq \rho} \hat{R}_S(h_{\theta+\epsilon}) + \gamma(\|\theta\|_2^2/\rho^2). \tag{40}$$

By Lemma A.3, the second term of R.H.S of Equation 39 can be bounded with the IPM between adversarial distribution and source distribution:

$$\text{IPM}_G(P_S^\Phi, P_T^\Phi) \leq M_\Phi \cdot \text{IPM}_G(P_S^\Phi, P_A^\Phi). \tag{41}$$

Denote $\alpha_\Phi = B_\Phi M_\Phi$, then $\alpha_\Phi > 0$ and

$$R_T(h_\theta) \leq \max_{\|\epsilon\| \leq \rho} \hat{R}_S(h_{\theta+\epsilon}) + \alpha_\Phi \text{IPM}(P_S^\Phi, P_A^\Phi) + \gamma(\|\theta\|_2^2/\rho^2). \tag{42}$$

$\qquad \square$

**Remark 1.** *Taking CRN (Bica et al., 2020b) as an example, we explain why the inconsistency between the train and test distributions weakens the effectiveness of learning balanced representations. Theorem 1 in CRN guarantees that the learned representations are independent of the current treatment assignment. Theorem 1 relies on the validity of Proposition 1 in the paper, and the core of its proof lies in the fact that the optimal prediction probabilities are given by:*

$$G_a^* = \arg\max_{G_a} \sum_{j=1}^K \int_{x'} \log(G_a^j(x')) P_j^\Phi(x') dx' \quad \text{subject to} \quad \sum_{j=1}^K G_a^j(x') = 1, \tag{43}$$

*where $x' = \Phi(\bar{h}_t)$. However, similar to our analysis in Section 3.2, we can conclude that $P_j^\Phi(x')$ during testing differs from that during training. Therefore, the $G_a^*$, which represents the optimal prediction probabilities learned based on the training set, may not retain its optimality on the test set. This leads to the inability to guarantee consistent conditional distributions $P(\Phi(\bar{\mathbf{H}}_t)|\mathbf{A}_t = \mathbf{a}_j)$ across different treatment assignments $\mathbf{a}_j$ during testing.*

# D DATASETS DESCRIPTION

## D.1 FS-TUMOR DATASET

In the study by (Geng et al., 2017), the authors model the volume of tumor growth for a period of $t + 1$ days following a cancer diagnosis using the Tumor Growth (TG) simulator, which generates one-dimensional outcomes. The simulator incorporates two treatment strategies: radiotherapy ($\mathbf{A}_t^r$) and chemotherapy ($\mathbf{A}_t^c$). In our implementation, we modify the treatment strategies to be continuous, ranging from 0 to 1, instead of binary. The treatments are applied in the model as follows: radiotherapy has an instantaneous effect $d(t)$ on the subsequent outcome when administered to a patient, while chemotherapy influences multiple future outcomes with a diminishing effect $C(t)$, described by the equation:

$$\mathbf{Y}_{t+1} = \left(1 + \rho \log\left(\frac{K}{\mathbf{Y}_t}\right) - \beta_C C(t) - (\alpha_r d(t) + \beta_r d(t)^2) + \epsilon_t\right)\mathbf{Y}_t, \tag{44}$$

where $\rho, K, \beta_C, \alpha_r, \beta_r$ are specified simulation parameters, and $\epsilon_t$ is the noise term, modeled as an independent sample from a normal distribution $N(0, 0.01^2)$. The individual patient response is characterized by the parameters $\beta_C, \alpha_r, \beta_r$, which are drawn from a mixture of three truncated normal distribution components. The mixture component indices serve as static covariates. For the precise parameter values, please consult the code implementation provided in the supplementary materials[2]. A biased treatment assignment introduces time-varying confounding for both treatments. The treatment assignment for radiotherapy ($\mathbf{A}_t^r$) and chemotherapy ($\mathbf{A}_t^c$) is sampled from beta distributions, expressed as:

$$\mathbf{A}_t^r, \mathbf{A}_t^c \sim \text{Beta}(2\sigma_t, 2 - 2\sigma_t), \tag{45}$$

where

$$\sigma_t = \sigma\left(\frac{\gamma}{D_{max}}\left(\bar{D}_{15}(\bar{\mathbf{Y}}_{t-1}) - D_{max}/2\right)\right), \tag{46}$$

with $\sigma(\cdot)$ representing a sigmoid activation function, $D_{max}$ the maximum tumor diameter, $\bar{D}_{15}(\bar{\mathbf{Y}}_{t-1})$ the average tumor diameter over the past 15 days, and $\gamma$ the confounding parameter. The parameter $\gamma$ allows for control over the degree of confounding. Specifically, when $\gamma = 0$, treatment assignments are completely randomized. Increasing $\gamma$ enhances the influence of time-varying confounding. In our adjustments, $d_t$ and $C_t$ are nonlinear functions of $\mathbf{A}_t^r$ and $\mathbf{A}_t^c$, respectively, fitted by cubic spline functions. Specifically, we have:

$$d(t) = 2\text{cs}_r(\mathbf{A}_t^r), \tag{47}$$
$$C(t) = 5\text{cs}_c(\mathbf{A}_t^c), \tag{48}$$

where $\text{cs}_r$ and $\text{cs}_c$ denote the cubic spline functions for radiotherapy and chemotherapy, respectively. This nonlinear functional setting introduces a more complex and realistic relationship between the treatment effects and the treatment intensities.

At each time step, for every patient in the test group, a set of counterfactual trajectories is generated based on $\tau$. For single-step predictions, all possible one-step-ahead counterfactual outcomes $Y_{t+1}$ are simulated, reflecting the tumor volume for each potential treatment combination. In the case of multi-step predictions, the number of potential outcomes for $Y_{t+2}, \ldots, Y_{t+\tau_{\max}}$ increases exponentially with the length of the forecast horizon $\tau_{\max}$.

Across different levels of confounding $\gamma$, we generate 1,000 patient trajectories for the training phase, 100 for validation, and another 100 for the testing phase. The duration of each trajectory is limited to a maximum of 60 time steps, with the understanding that some patients may have shorter trajectories due to recovery or demise.

In line with previous studies (Bica et al., 2020b; Melnychuk et al., 2022), we compute the normalized RMSE, which is scaled relative to the maximum tumor volume $V_{\max} = 1150$ cubic centimeters.

---

[2]Please refer to our supplementary materials.

## D.2 CISD DATASET

Drawing on the methodology outlined in (Wang et al., 2024), we construct a synthetic dataset featuring continuous interventions modeled through an autoregressive process, where the treatment variable $\mathbf{A}_t$ ranges within $[0, 1]$. The time series is iteratively generated following the detailed steps below.

The generation of the treatment variable $\mathbf{A}_t$ at each time $t$ involves manipulating historical covariate data. The average of these historical covariates, denoted as $\mathbf{X}_m$, is calculated over time:

$$\mathbf{X}_m = \frac{1}{w} \sum_{i=1}^{w} \mathbf{X}_{t-i}, \tag{49}$$

where $w$ is the window size of past time steps influencing the current value.

This mean covariate vector $\mathbf{X}_m$ undergoes multiple non-linear transformations coupled with noise addition, forming a decision variable:

$$d_t = \sin(2\pi \mathbf{X}_m^1) + \cos(2\pi \mathbf{X}_m^2) \times \mathbf{X}_m^5 + \max(\mathbf{X}_m^3, \mathbf{X}_m^4) + N(0, \sigma_a^2), \tag{50}$$

where $\sigma_a$ denotes the noise scale for the treatment variable, and superscripts indicate specific elements within $\mathbf{X}_m$. From the decision variable $d_t$, a probability $p_t$ is derived using the sigmoid function:

$$p_t = \frac{1}{1 + \exp(-d_t)}. \tag{51}$$

The treatment variable $\mathbf{A}_t$ is then sampled from a Beta distribution, influenced by $p_t$ and scaled by $\gamma$, formalized as:

$$\gamma_1 = 1 + \gamma \times p_t, \quad \gamma_2 = 1 + \gamma \times (1 - p_t), \tag{52}$$
$$\mathbf{A}_t \sim \text{Beta}(\gamma_1, \gamma_2). \tag{53}$$

This setup ensures that $\mathbf{A}_t$ encapsulates the effects of historical covariate dynamics through non-linear, noise-induced transformations, and represents treatment effects stochastically with probabilistic modeling using the Beta distribution.

For each treatment variable $\mathbf{A}_t$, a transformed treatment array, $T(\mathbf{A}_t)$, is first obtained through non-linear transformations. This array is then subjected to a specific masking procedure to derive a transformed matrix $\mathbf{A}_{\text{matrix}}$.

The covariates at time $t$, $\mathbf{X}_t$, are generated by blending $\mathbf{X}_m$ with $\mathbf{A}_{\text{matrix}}$ and adding Gaussian noise:

$$\mathbf{X}_t = \mathbf{X}_m \times \mathbf{A}_{\text{matrix}} + N(0, \sigma_x^2), \tag{54}$$

where $\sigma_x$ is the noise scale for $\mathbf{X}$.

The outcome at time $t$, $\mathbf{Y}_t$, is then produced by integrating $\mathbf{X}_m$ with $\mathbf{A}_t$ and incorporating Gaussian noise:

$$\mathbf{Y}_t = \cos(2\pi \mathbf{A}_t) \times \mathbf{X}_m^1 + \mathbf{A}_t^2 \times \mathbf{X}_m^4 + \sin(2\pi \mathbf{A}_t) \times \mathbf{X}_m^6 + \exp(\mathbf{A}_t) \times \mathbf{X}_m^3 + N(0, \sigma_y^2), \tag{55}$$

where $\sigma_y$ is the noise scale for $\mathbf{Y}$.

For immediate future predictions, we randomly select five interventions $\mathbf{A}_t$ from a uniform distribution $U(0, 1)$ to compute counterfactual outcomes. When predictions extend over multiple steps, defined by $\tau_{\text{max}}$, we generate a diverse set of trajectories for each patient at every step, corresponding to the projection horizon.

## E   HYPERPARAMETER TUNING

Hyperparameter optimization was conducted for all baseline models including RMSN, CRN, G-Net, CT, and ACTIN through a random grid search. The ranges for these random searches for RMSN, CRN, G-Net, and CT are detailed in Tables 2, 3, 4, and 5 respectively. The search space for ACTIN is outlined in Table 6.

For ACTIN, hyperparameter optimization was initially performed for two distinct base models, TCN and LSTM. However, after evaluation, the TCN model was selected as the sole base model for ACTIN due to its superior performance. The optimization for the TCN model within ACTIN specifically involved adjustments to channel sizes, dilation factors, and kernel sizes, as specified in Table 6. Our model, TACIN, also underwent a similar process of hyperparameter optimization to ensure optimal performance.

Table 2: The hyperparameter tuning ranges for RMSN are tailored for different datasets. In the table, $N_{Pt}$, $N_{Ph}$, $N_E$, and $N_D$ represent the Propensity treatment network, Propensity history network, Encoder, and Decoder sub-models, respectively. It is generally presumed that the hyperparameter ranges apply uniformly across all sub-models unless specified otherwise.

| Hyperparameter | Range (FS-Tumor) | Range (CISD) |
|---|---|---|
| LSTM layers | 1 | 1, 2 |
| Learning rate | 0.01, 0.001, 0.0001 | 0.01, 0.001, 0.0001 |
| Minibatch size ($N_{Pt}, N_{Ph}, N_E$) | 64, 128, 256 | 64, 128, 256 |
| Minibatch size ($N_D$) | 256, 512, 1024 | 256, 512, 1024 |
| LSTM hidden units ($N_{Pt}$) | 8, 12, 16 | 4, 8, 12 |
| LSTM hidden units ($N_{Ph}, N_E$) | 8, 12, 16 | 4, 8, 12 |
| LSTM hidden units ($N_D$) | 16, 32, 64 | 4, 8, 12 |
| LSTM dropout rate | 0.1, 0.2, 0.3, 0.4, 0.5 | 0.1, 0.2, 0.3, 0.4, 0.5 |
| Max gradient norm ($N_{Pt}, N_{Ph}, N_E$) | 0.5, 1.0, 2.0 | 0.5, 1.0, 2.0 |
| Max gradient norm ($N_D$) | 0.5, 1.0, 2.0, 4.0 | 0.5, 1.0, 2.0, 4.0 |
| Random search iterations | 30 | 30 |
| Number of epochs | 150 | 150 |

Table 3: The specified hyperparameter tuning ranges for CRN vary across different datasets. In this context, $N_E$ and $N_D$ represent the Encoder and Decoder sub-models, respectively. It is generally assumed that these sub-models follow the same hyperparameter ranges unless noted otherwise.

| Hyperparameter | Range (FS-Tumor) | Range (CISD) |
|---|---|---|
| LSTM layers | 1 | 1, 2 |
| Learning rate | 0.01, 0.001, 0.0001 | 0.01, 0.001, 0.0001 |
| Minibatch size ($N_E$) | 64, 128, 256 | 64, 128, 256 |
| Minibatch size ($N_D$) | 256, 512, 1024 | 256, 512, 1024 |
| LSTM hidden units ($N_E$) | 3, 6, 12, 18, 24 | 8, 16, 24, 32 |
| LSTM hidden units ($N_D$) | $d_r^e$ | $d_r^e$ |
| BR size $d_r^e$ ($N_E$) | 3, 6, 12, 18, 24 | 3, 6, 9, 16, 32 |
| BR size $d_r^d$ ($N_D$) | 3, 6, 12, 18, 24 | 6, 12, 18, 24, 48 |
| FC hidden units ($N_E$) | 6, 9, 12, 24, 48, 96 | 8, 12, 16, 24, 32 |
| FC hidden units ($N_D$) | 6, 9, 12, 24, 48, 96 | 8, 12, 16, 24, 32 |
| LSTM dropout rate | 0.1, 0.2, 0.3, 0.4, 0.5 | 0.1, 0.2, 0.3, 0.4, 0.5 |
| Random search iterations | 30 | 30 |
| Number of epochs | 150 | 150 |

# F  ADDITIONAL RESULTS

In this section, we provide the complete experimental results on the FS-Tumor dataset, with $\gamma$ ranging from 0 to 4 and $\tau$ from 1 to 6, as shown in Table 8. The results demonstrate that TACIN

Table 4: The specified ranges for hyperparameter tuning of G-Net vary across different datasets.

| Hyperparameter | Range (FS-Tumor) | Range (CISD) |
|---|---|---|
| LSTM layers | 1 | 1, 2 |
| Learning rate | 0.01, 0.001, 0.0001 | 0.01, 0.001, 0.0001 |
| Minibatch size | 64, 128, 256 | 64, 128, 256 |
| LSTM hidden units | 12, 18, 24 | 8, 16, 32 |
| LSTM output size $d_o$ | 12, 18, 24 | 3, 6, 12, 24 |
| FC hidden units | 6, 12, 18, 24 | 12, 24, 48 |
| LSTM dropout rate | 0.1, 0.2, 0.3, 0.4, 0.5 | 0.1, 0.2, 0.3, 0.4, 0.5 |
| Random search iterations | 30 | 30 |
| Number of epochs | 150 | 150 |

Table 5: Hyperparameter tuning ranges for CT across different datasets.

| Hyperparameter | Range (FS-Tumor) | Range (CISD) |
|---|---|---|
| Transformer blocks | 1 | 1 |
| Learning rate | 0.01, 0.001, 0.0001 | 0.01, 0.001, 0.0001 |
| Minibatch size | 64, 128, 256 | 64, 128, 256 |
| Attention heads | 2 | 2 |
| Transformer units | 8, 12, 16, 32 | 8, 16, 32 |
| BR size $d_r$ | 8, 12, 16, 32 | 8, 16, 32 |
| FC hidden units | 8, 12, 16, 32, 64 | 8, 16, 32 |
| Sequential dropout rate | 0.1, 0.2, 0.3, 0.4, 0.5 | 0.1, 0.2, 0.3, 0.4, 0.5 |
| Max positional encoding | 15 | 15 |
| Random search iterations | 30 | 30 |
| Number of epochs | 150 | 150 |

consistently outperforms the baselines across different $\gamma$ values and prediction horizons. In particular, TACIN achieves the lowest RMSE values for $\gamma$ values of 2, 3, and 4, indicating its superior performance in estimating the counterfactual outcomes. The results further confirm the effectiveness of TACIN in handling time-varying confounding bias and improving the accuracy of causal effect estimation in longitudinal data analysis.

Table 6: Specified ranges for hyperparameter tuning of ACTIN across various datasets.

| Hyperparameter | Range (FS-Tumor) | Range (CISD) |
|---|---|---|
| Linear transformation size | 4, 8, 16 | 16, 32, 64 |
| Learning rate $l$ | 0.01, 0.001, 0.0001 | 0.01, 0.001, 0.0001 |
| Learning rate $l_{\mathcal{D}}$ | 0.001, 0.0002, 0.0001 | 0.001, 0.0002, 0.0001 |
| Minibatch size | 64, 128, 256 | 64, 128, 256 |
| BR size $d_r$ | 8, 12, 16, 32 | 8, 16, 32 |
| TCN-based Kernel sizes | 2, 3 | 2, 3 |
| Dilation factors | 2, 3 | 2, 3 |
| Channel size $d_c$ | $d_r$ | $d_r$ |
| FC hidden units | 16, 32, 64 | 16, 32, 64 |
| Dropout rate | 0.1, 0.2, 0.3 | 0.1, 0.2, 0.3 |
| Random search iterations | 30 | 30 |
| Number of epochs | 150 | 150 |

Table 7: Specified ranges for hyperparameter tuning of TACIN across various datasets.

| Hyperparameter | Range (FS-Tumor) | Range (CISD) |
|---|---|---|
| Learning rate $l$ | 0.01, 0.001, 0.0001 | 0.01, 0.001, 0.0001 |
| Minibatch size | 64, 128, 256 | 64, 128, 256 |
| Representation size $d_r$ | 8, 16, 32 | 4, 8, 16 |
| RBF | multiRBF, guassianRBF | multiRBF, guassianRBF |
| IFC Kernel sizes | 2, 3 | 2, 3 |
| Dilation factors | 3 | 3 |
| Channel size $d_c$ | $d_r$ | $d_r$ |
| FC hidden units | 16, 32, 64 | 16, 32, 64 |
| Dropout rate | 0, 0.1, 0.2, 0.3 | 0, 0.1, 0.2, 0.3 |
| Random search iterations | 30 | 30 |
| Number of epochs | 150 | 150 |

Table 8: The one-step-ahead and multi-step-ahead prediction results for the FS-Tumor dataset. Shown: RMSE as mean $\pm$ standard deviation over ten runs.

| | | $\tau = 1$ | $\tau = 2$ | $\tau = 3$ | $\tau = 4$ | $\tau = 5$ | $\tau = 6$ |
|---|---|---|---|---|---|---|---|
| $\gamma = 0$ | RMSN | $0.72 \pm 0.24$ | $0.86 \pm 0.23$ | $0.88 \pm 0.23$ | $0.84 \pm 0.23$ | $0.78 \pm 0.23$ | $0.71 \pm 0.22$ |
| | CRN | $\mathbf{0.67 \pm 0.18}$ | $0.82 \pm 0.21$ | $0.85 \pm 0.22$ | $0.82 \pm 0.22$ | $0.76 \pm 0.21$ | $0.68 \pm 0.19$ |
| | G-Net | $0.70 \pm 0.21$ | $0.94 \pm 0.21$ | $1.02 \pm 0.24$ | $0.97 \pm 0.23$ | $0.90 \pm 0.23$ | $0.82 \pm 0.21$ |
| | CT | $0.69 \pm 0.17$ | $0.82 \pm 0.19$ | $0.84 \pm 0.21$ | $0.80 \pm 0.20$ | $0.74 \pm 0.19$ | $0.67 \pm 0.18$ |
| | ACTIN | $0.69 \pm 0.24$ | $0.83 \pm 0.20$ | $0.85 \pm 0.21$ | $0.79 \pm 0.20$ | $0.73 \pm 0.20$ | $0.66 \pm 0.19$ |
| | TACIN | $0.68 \pm 0.17$ | $\mathbf{0.79 \pm 0.15}$ | $\mathbf{0.81 \pm 0.17}$ | $\mathbf{0.77 \pm 0.18}$ | $\mathbf{0.71 \pm 0.17}$ | $\mathbf{0.66 \pm 0.15}$ |
| $\gamma = 1$ | RMSN | $0.69 \pm 0.13$ | $0.86 \pm 0.27$ | $0.90 \pm 0.28$ | $0.87 \pm 0.29$ | $0.82 \pm 0.30$ | $0.75 \pm 0.31$ |
| | CRN | $0.78 \pm 0.17$ | $0.93 \pm 0.30$ | $0.90 \pm 0.28$ | $0.82 \pm 0.25$ | $0.75 \pm 0.23$ | $\mathbf{0.67 \pm 0.23}$ |
| | G-Net | $\mathbf{0.68 \pm 0.15}$ | $0.94 \pm 0.28$ | $1.00 \pm 0.28$ | $0.96 \pm 0.28$ | $0.89 \pm 0.26$ | $0.82 \pm 0.27$ |
| | CT | $0.71 \pm 0.12$ | $0.88 \pm 0.30$ | $0.88 \pm 0.29$ | $0.82 \pm 0.28$ | $0.76 \pm 0.26$ | $0.70 \pm 0.25$ |
| | ACTIN | $0.70 \pm 0.14$ | $0.86 \pm 0.30$ | $0.86 \pm 0.28$ | $\mathbf{0.81 \pm 0.26}$ | $0.76 \pm 0.24$ | $0.70 \pm 0.24$ |
| | TACIN | $0.69 \pm 0.17$ | $\mathbf{0.82 \pm 0.29}$ | $\mathbf{0.85 \pm 0.28}$ | $0.81 \pm 0.29$ | $\mathbf{0.74 \pm 0.26}$ | $0.67 \pm 0.25$ |
| $\gamma = 2$ | RMSN | $0.85 \pm 0.12$ | $0.97 \pm 0.27$ | $1.03 \pm 0.33$ | $1.00 \pm 0.36$ | $0.95 \pm 0.36$ | $0.89 \pm 0.39$ |
| | CRN | $0.86 \pm 0.18$ | $0.91 \pm 0.26$ | $0.97 \pm 0.31$ | $0.92 \pm 0.34$ | $0.86 \pm 0.32$ | $0.80 \pm 0.34$ |
| | G-Net | $0.81 \pm 0.18$ | $0.98 \pm 0.28$ | $1.07 \pm 0.33$ | $1.03 \pm 0.34$ | $0.95 \pm 0.32$ | $0.90 \pm 0.34$ |
| | CT | $0.87 \pm 0.23$ | $0.91 \pm 0.24$ | $0.99 \pm 0.30$ | $0.94 \pm 0.31$ | $0.91 \pm 0.31$ | $0.86 \pm 0.31$ |
| | ACTIN | $0.79 \pm 0.21$ | $0.87 \pm 0.24$ | $0.93 \pm 0.29$ | $0.89 \pm 0.31$ | $0.83 \pm 0.29$ | $0.77 \pm 0.30$ |
| | TACIN | $\mathbf{0.78 \pm 0.16}$ | $\mathbf{0.85 \pm 0.24}$ | $\mathbf{0.88 \pm 0.28}$ | $\mathbf{0.84 \pm 0.29}$ | $\mathbf{0.77 \pm 0.27}$ | $\mathbf{0.71 \pm 0.28}$ |
| $\gamma = 3$ | RMSN | $1.09 \pm 0.40$ | $1.27 \pm 0.39$ | $1.34 \pm 0.40$ | $1.29 \pm 0.46$ | $1.27 \pm 0.59$ | $1.21 \pm 0.70$ |
| | CRN | $1.09 \pm 0.44$ | $1.25 \pm 0.41$ | $1.33 \pm 0.47$ | $1.30 \pm 0.43$ | $1.26 \pm 0.44$ | $1.16 \pm 0.41$ |
| | G-Net | $1.13 \pm 0.42$ | $1.42 \pm 0.48$ | $1.50 \pm 0.50$ | $1.38 \pm 0.48$ | $1.26 \pm 0.46$ | $1.14 \pm 0.45$ |
| | CT | $1.17 \pm 0.45$ | $1.20 \pm 0.39$ | $1.31 \pm 0.45$ | $1.34 \pm 0.46$ | $1.27 \pm 0.41$ | $1.17 \pm 0.40$ |
| | ACTIN | $\mathbf{1.08 \pm 0.39}$ | $1.20 \pm 0.37$ | $1.28 \pm 0.45$ | $1.27 \pm 0.46$ | $1.20 \pm 0.46$ | $1.10 \pm 0.44$ |
| | TACIN | $1.09 \pm 0.43$ | $\mathbf{1.09 \pm 0.30}$ | $\mathbf{1.19 \pm 0.38}$ | $\mathbf{1.16 \pm 0.39}$ | $\mathbf{1.08 \pm 0.34}$ | $\mathbf{0.99 \pm 0.32}$ |
| $\gamma = 4$ | RMSN | $1.52 \pm 0.55$ | $1.94 \pm 1.06$ | $2.10 \pm 1.30$ | $1.95 \pm 1.22$ | $1.78 \pm 1.09$ | $1.57 \pm 0.97$ |
| | CRN | $1.61 \pm 0.63$ | $2.11 \pm 1.19$ | $2.16 \pm 1.30$ | $2.07 \pm 1.24$ | $1.91 \pm 1.17$ | $1.69 \pm 1.01$ |
| | G-Net | $1.47 \pm 0.52$ | $2.08 \pm 1.11$ | $2.27 \pm 1.29$ | $2.16 \pm 1.25$ | $2.01 \pm 1.20$ | $1.87 \pm 1.18$ |
| | CT | $1.66 \pm 0.58$ | $2.22 \pm 1.09$ | $2.12 \pm 1.00$ | $1.99 \pm 0.90$ | $1.85 \pm 0.83$ | $1.74 \pm 0.81$ |
| | ACTIN | $1.58 \pm 0.47$ | $1.86 \pm 0.92$ | $1.90 \pm 1.02$ | $1.79 \pm 0.97$ | $1.69 \pm 0.94$ | $1.58 \pm 0.89$ |
| | TACIN | $\mathbf{1.31 \pm 0.52}$ | $\mathbf{1.54 \pm 0.76}$ | $\mathbf{1.69 \pm 0.90}$ | $\mathbf{1.62 \pm 0.88}$ | $\mathbf{1.55 \pm 0.87}$ | $\mathbf{1.39 \pm 0.81}$ |

