# OpenReview forum: "Temporal Adaptive Convolutional Intervention Network for Counterfactual Estimation: A Domain Generalization Perspective"
_ICLR.cc/2025/Conference — ICLR 2025 Conference Withdrawn Submission_

### Official Review · Reviewer_6TmC · 2024-10-18

**Soundness:** 1
**Presentation:** 3
**Contribution:** 1
**Rating:** 3
**Confidence:** 4

**Summary:**

The paper develops a neural method for estimating counterfactual outcomes for longitudinal data. Specifically, the authors use balancing and convolutional kernels in their neural network. The authors emphasize that they capture temporal dependencies of treatments with their method and frame the problem of estimating counterfactual outcomes as a domain generalization problem.

**Strengths:**

- **Readability:** The paper is very well written and easy to read.
- **Implementation details:** I highly value the details on hyperparameter tuning and on the datasets for reproducibility.

**Weaknesses:**

- **Confounding bias:** The paper completely misses the point in defining what confounding bias actually means in the longitudinal setting. In section 3.2, the authors unfortunately fail to properly introduce this and, instead, use a very heuristic intuition. For clarification on this matter, I highly recommend the papers by Robins (which are actually cited in the paper ...). There are several approaches to adjust for time-varying confounders, such as regression adjustments and propensity adjustments (e.g., marginal structural models, G-computation, inverse propensity weighting, ...).

- **Objective function:** Equation (1) correctly defines the objective. However, it remains completely unclear how the counterfactual outcome for a treatment intervention can be identified under the three assumptions -- what is the causal estimand? In fact, this is only possible through proper adjustments (see previous bullet point).

- **Balancing:** Balancing does **not** reduce confounding bias. Instead, it reduces estimation variance. Claiming that balancing is an adjustment for time-varying confounding is **not valid**. While previous works do show that balancing helps improving estimation error, this likely stems from reduced estimation variance. For proper adjustments in the time-varying setting, I highly recommend the aforementioned papers by Robins. For a better understanding of balancing, I highly recommend the original work by Shalit.

- **Convolution / novelty:** Using convolutions is not novel for neural networks that process time-varying data. Further, I do not see the benefit of this specific approach for counterfactual estimation.

- **Generalization bound:** The proposed generalization bound has -- from my understanding -- no practical relevance, as there is no way to bound the constant $M_\Phi$. Further, as mentioned above, minimizing the IPM is not an adjustment for time-varying confounding.

- **Motivation for domain generalization:** Lines 219-225 do not make sense to me. It is correct that we do not know the distribution of the counterfactual outcomes. But this holds for **any** regression task (which is why we are interested in the mean and perform a regression...). There is no clear motivation why counterfactual outcome regression is a domain generalization problem.

- **Complex treatment dependencies:** One of the main motivations for this paper is to capture "capture complex temporal treatment interactions". I do not see how any of the proposed methodology approaches this (on top, there is not even a proper formalization what "complex treatment dependencies" means).

**Questions:**

- What is the authors' understanding of confounding bias?

- Why are the proposed convolutions helpful for this specific task? And what is the novelty?

- What is the practical relevance of the generalization bound?

- Can the authors clarify the motivation for domain generalization?

- How are "complex treatment dependencies" captured?

---

### Official Review · Reviewer_sXvk · 2024-11-03

**Soundness:** 4
**Presentation:** 2
**Contribution:** 3
**Rating:** 6
**Confidence:** 4

**Summary:**

This study derives a generalization bound on counterfactual risk using Sharpness-Aware Minimization and the previous result of generalization bound with Integral Probability Metric. To address the limitation in directly applying the previous bound to the current problem setup, the authors derive a bound with Adversarial distribution. Based on the theoretical results, the authors propose Temporal Adaptive Convolutional Intervention Network (TACIN) to estimate counterfactual outcomes over time.

**Strengths:**

The generalization bound is based on the previous known results and mathematically sound valid.

The proposed method is motivated by and based on the theoretical results of the generalization bound.

**Weaknesses:**

The authors should consider providing a more detailed rationale for selecting the WaveNet architecture (van den Oord et al., 2016) as the backbone. Given that the module for generating learned representations from input data could be flexibly designed with other sequence-to-sequence architectures, such as RNNs or Transformers, the justification for this specific choice needs clarification. Additionally, a previous study, ACTIN (Wang et al., 2024), which shares architectural similarities, conducted an ablation study by replacing WaveNet with LSTM. Performing a similar ablation study would greatly strengthen the authors' case by demonstrating the robustness and validity of their architectural decisions.

In equation (2), the authors used the intervention function to capture nonlinear interactions between variables. The reviewer is concerned that F(t) can still learn similar interactions (through equations (2) and (14)) between treatments without the interaction function proposed if z includes treatments. To address this concern, an ablation study that evaluates the impact of the proposed interaction function would be highly informative. Such a study would help validate the necessity and effectiveness of incorporating the interaction function into the model.

The author is concerned if the current ablation study derives the conclusion based on the limited observation since the experiment was only conducted on the tumor data with gamma=4. Similar experiments on the other dataset and other settings will help generalize the author’s claim.

**Questions:**

Typos in section 4.3 line 470-472 “strategie”

---

### Official Review · Reviewer_tscG · 2024-11-04

**Soundness:** 2
**Presentation:** 3
**Contribution:** 2
**Rating:** 3
**Confidence:** 4

**Summary:**

The authors propose a method for inferring treatment effects in temporal settings under confounding through counterfactual estimation. Specifically, the model the authors introduce known as Temporal Adaptive Convolutional Intervention Network (TACIN) employs a convolutional kernel over time to capture temporal treatment interactions. Radial basis functions are then used to model nonlinear interactions between treatments. Finally, the authors derive a generalization bound for the model's performance under various conditions.

**Strengths:**

I think the problem of reliably inferring counterfactuals is a tricky one but one that many view as critical in order to infer the treatment effect. Few such as Philip Dawid oppose the view of counterfactuals (see for instance his work on causal inference without counterfactuals https://www.jstor.org/stable/2669377). In that sense, I think the authors address a very relevant problem. I also think the fact that the authors consider the role of time in the process and try to capture temporal interactions is important.  Overall the paper is written well and the authors try to analyse and provide some theoretical justifications for their approach along with the empirical analysis. The experiments also show the performance of the approach in continuous intervention settings.

**Weaknesses:**

The paper tries to do a lot but does not clarify some important details that are crucial for the feasability and plausibility of the approach in practice. Specifically,

1) The authors need to make very explicit what their assumptions are. In order to perform counterfactual estimation, there needs to be an underlying assumed causal graph. The authors never show us what this graph is.

2) The paper makes very strong assumptions about sequential ignorability and sequential overlap that simplify the problem considerably and limit the novelty of the approach. If the authors assume sequential ignorability, the problem of inferring treatment effects simply reduces to representation learning in sequential settings where the representation should capture all the relevant information that is observed. In the case where confounders are known, estimating treatment effects simply requires conditioning on the known confounders. As a result, any reasonable representation learning method should suffice. The difficulty lies in cases where confounders may be hidden and may introduce long-term biases in the process.

3)  The authors state that prior works assume the distributions of histories during training and testing remains the same which may not occur in real life. The proposed approach is supposed to work in settings where domain generalization is necessary/where these distributions differ. In the experimental analysis, I would have liked to have seen visualisations of the different distributions at train and test time so see how significantly these differ and some visualisation of how the approach manages to generalize across both of them. I would have also liked to have seen some analysis on the performance of the approach as this difference between train and test distributions becomes bigger.

4) The authors only test their approach on two synthetic datasets where the counterfactuals are available. It is clear that temporal datasets containing counterfactuals are not easy to find however, I wonder if the authors could have designed their own synthetic data set through simulating different processes (with varying degrees of distribution shift between train and test) that could allow us to inspect various aspects of the performance.

**Questions:**

1) Can you provide details of the assumptions made and show me what the assumed graph looks like

2) How are these assumptions plausible and why do you need such a complex method for learning these representations as opposed to using standard methods for causal representation learning especially considering you make the assumption of sequential ignorability.

3) How does the proposed method compare to other methods that capture temporal dependencies and estimate treatment effects such as https://arxiv.org/pdf/1704.02038 or https://arxiv.org/pdf/1703.10651. Unfortunately there are no baselines provided which makes it hard to gauge performance or justify claims.

4) I would have liked to have seen visualisations of the different distributions at train and test time so see how significantly these differ and some visualisation of how the approach manages to generalize across both of them. I would have also liked to have seen some analysis on the performance of the approach as this difference between train and test distributions becomes bigger.

5) I wonder if the authors could have designed their own synthetic data set through simulating different processes (with varying degrees of distribution shift between train and test) that could allow us to inspect various aspects of the performance.

---

### Official Review · Reviewer_vsUX · 2024-11-04

**Soundness:** 1
**Presentation:** 2
**Contribution:** 1
**Rating:** 3
**Confidence:** 3

**Summary:**

The paper suggests a new method to accurately estimate time-varying treatment effects and counterfactual outcomes, namely Temporal Adaptive Convolutional Intervention Network (TACIN). In the TACIN, the authors employed convolutional neural network layers based on radial basis functions, specifically tailored for interventional predictions. Additionally, the authors suggested using a new adversarial objective to build balanced representations, where they extended an existing approach of integral probability metrics (IPMs) to sharpness-aware minimisation (SAM) via adversarial distributions. This was done to achieve a better predictive generalisation of different counterfactual treatments.  In the end, several synthetic and semi-synthetic experiments were provided to show the effectiveness of the TACIN.

**Strengths:**

The work addresses a challenging and important task: counterfactual prediction over time. The authors explored an interesting connection between generalisation bounds via IMP, developed for counterfactual prediction in the cross-sectional setting, and generalisation bounds for domain adaptation with SAM via adversarial distributions.

**Weaknesses:**

I have several major concerns regarding the soundness and correctness of the proposed method:
- Although I appreciate the authors’ theoretical contributions (Lemmas 1-3, Theorem 1) on the generalisation bounds for counterfactual prediction, I do not see how they can be practically incorporated into a training objective, e.g., Eq. 17. Specifically, as Lemma 3 is true for any **fixed** representation $\Phi$, the constant $M_\Phi$ may change when  $\Phi$ is learnable.
- Additionally, it is unclear whether the invertibility of the representation is required for the TACIN, as in the original work of [1]. For example, without invertibility, the representations may be confounded if too much balancing is enforced [2].
- No discussion is provided on the counterfactual identification, namely, how to infer several-step-ahead counterfactual outcomes from observational data given the causal assumptions (Appendix B). It seems to me like the authors are not doing a proper time-varying adjustment (e.g., g-computation (G-Net) or inverse propensity weighting (RMSN)).

Also, I have several concerns regarding the empirical evaluation of the methods:
- I wonder what is the gain of the proposed adversarial loss over the vanilla IPM weighting (I encourage authors to provide such a comparison).
- The results of the ablation study in Fig. 4 do not look convincing. That is the performance of the TACIN with and without the proposed adversarial objectives looks almost equivalent on the box plots in Fig. 4.

Overall, in my opinion, the paper's contribution is only marginal compared with the existing IPM balancing methods for counterfactual prediction.

References:
- [1] Shalit, Uri, Fredrik D. Johansson, and David Sontag. "Estimating individual treatment effect: generalization bounds and algorithms." International conference on machine learning. PMLR, 2017.
- [2] Melnychuk, Valentyn, Dennis Frauen, and Stefan Feuerriegel. "Bounds on Representation-Induced Confounding Bias for Treatment Effect Estimation." The Twelfth International Conference on Learning Representations. 2024

**Questions:**

- What is the reason behind modelling covariates (Eq. 16), if the main task is the counterfactual outcomes prediction?
- How were the hyperparameters $\lambda$ from Eq. 17 chosen?
- I don’t understand how Eq. 7 relates to the initial target quantity, i.e., counterfactual outcomes under fixed hard interventions (Eq.1). Shouldn’t P_test be a Dirac delta distribution?

---

### Note · Authors · 2024-11-13

I have read and agree with the venue's withdrawal policy on behalf of myself and my co-authors.